Three new species of Bertolonia (Melastomataceae) from Espírito Santo, Brazil

Bacci Lucas F. lucasfbacci@gmail.com 1
Amorim André M. 2 3
Goldenberg Renato 4
1 Programa de Pós Graduação em Biologia Vegetal, Instituto de Biologia, Universidade Estadual de Campinas, Rua Monteiro Lobato 255, Cidade Universitária Zeferino Vaz, Barão Geraldo, Universidade Estadual de Campinas , Campinas , São Paulo , Brazil
2 Departamento de Ciências Biológicas, Universidade Estadual de Santa Cruz , Ilhéus , Bahia , Brazil
3 Centro de Pesquisas do Cacau, Herbário CEPEC , Itabuna , Bahia , Brazil
4 Departamento de Botânica, Centro Politécnico, Universidade Federal do Paraná , Curitiba , Paraná , Brazil
Pie Marcio
Electronic publication date: 2016 Dec 22
Publication date: 2016
Volume: 4
Electronic Location ID: e2822
Received 2016 Oct 7; Accepted 2016 Nov 22
Copyright: ©2016 Bacci et al.
Copyright year: 2016
Copyright holder: Bacci et al.
License: This is an open access article distributed under the terms of the Creative Commons Attribution License, which permits unrestricted use, distribution, reproduction and adaptation in any medium and for any purpose provided that it is properly attributed. For attribution, the original author(s), title, publication source (PeerJ) and either DOI or URL of the article must be cited.
License URL: https://creativecommons.org/licenses/by/4.0/

Keywords: Atlantic forest, Bertolonieae, Endemism, Taxonomy

Funding: CAPES PhD grant for the Programa de Pòs-Graduação em Biologia Vegetal, UNICAMP CNPq “Produtividade em Pesquisa” # 310717/2015-9 # 306852/2013-6 Fundação O Boticário de Proteção a Natureza # 200720074 National Science Foundation DEB-818399 DEB-1343612 LFB received financial support from a CAPES PhD grant, AMA and RG from CNPq “Produtividade em Pesquisa,” grants # 310717/2015-9 and # 306852/2013-6, respectively. The fieldwork of AMA and RG in Espirito Santo State was made possible by support from the Fundação O Boticário de Proteção a Natureza (grant # 200720074). LFB’s fieldwork was partially funded by the National Science Foundation through grants to FA Michelangeli (DEB-818399 and DEB-1343612) The funders had no role in study design, data collection and analysis, decision to publish, or preparation of the manuscript.

==============================
We describe and illustrate three new species of Bertolonia, all endemic to the state of Espírito Santo, Brazil. Bertolonia duasbocaensis and B. macrocalyx occur close to each other, in the municipalities of Cariacica and Viana. Bertolonia ruschiana has a wider distribution, occurring in the municipalities of Santa Leopoldina, Santa Maria de Jetibá and Santa Teresa. The first two species are classified as critically endangered (CR), and the latter as endangered (EN). We also present an identification key for the species of Bertolonia that occur in Espírito Santo.

Introduction

Bertolonia Raddi is a Neotropical genus of Melastomataceae endemic to the Atlantic Forest biome (Baumgratz, 1990; Bacci et al., 2016). The genus was previously classified as a member of tribe Bertolonieae, together with other genera of Melastomataceae that are also herbs with obtriquetrous capsular fruits, like Monolena Triana ex Bent. & Hook. f., Salpinga Mart. ex DC. and Triolena Naudin (Clausing & Renner, 2001). Nevertheless, the tribe is not monophyletic, and Monolena, Salpinga and Triolena actually belong to different evolutionary lineages (Goldenberg et al., 2012). Inside the Atlantic Forest biome, the species of Bertolonia occur mostly in the Atlantic Rainforest (“Floresta Ombrófila Densa Submontana/Montana,” following the official Brazilian system—(Veloso, Rangel-Filho & Lima, 1991). It has 22 taxa, including Bertolonia margaritacea Naudin (until recently recognized as Salpinga margaritacea (Naudin) DC.; see Bacci et al., 2016; Goldenberg, Bacci & Bochorny, 2016). Bertolonia species are small, perennial herbs that inhabit moist and shaded areas. They grow on forest litter, directly on rocks or on shallow soils, on decaying logs or as epiphytes on the base of tree trunks (frequently tree ferns). The plants are usually glandulose-punctate or also with other types of trichomes, rarely glabrous (Baumgratz, 1990). The inflorescences are scorpioid cymes and the fruits are obtriquetrous (Baumgratz, 1990), bertolonidium-type capsules (following Baumgratz, 1983–1985).

Four new species of Bertolonia have been described in the last five years (Baumgratz, Amorim & Jardim, 2011; Bacci et al., 2016; Silva-Gonçalves et al., 2016), three from the state of Bahia and one from Rio de Janeiro. According to “Flora do Brasil” (Baumgratz, 2016), there should be only five species of Bertolonia occurring in the state of Espírito Santo: Bertolonia formosa Brade, B. foveolata Brade, B. maculata DC., B. sanguinea Saldanha ex Cogn. and B. wurdackiana Baumgratz. Since the specimens that have been mentioned as B. sanguinea actually belong to one of the three new species described here, the total number for Bertolonia in Espírito Santo should rise to seven species. Espírito Santo was originally and almost entirely covered by the Atlantic Forest Biome (Fundação SOS Mata Atlântica & INPE, 1993). Nowadays, only about 10% of the original vegetation remains (Fundação SOS Mata Atlântica & INPE, 2014). Despite the small remaining forested area and the relatively homogenous vegetation, Espírito Santo has several rare, endemic and threatened species of both fauna and flora, and can be considered a biodiversity hotspot (Dutra, Alves-Araújo & Carrijo, 2015).

We describe here three new species of Bertolonia that are endemic to Espírito Santo, Brazil. We also provide an identification key for the species that occur in Espírito Santo, comparisons with closely related species, information about distribution and conservation status, and photos of living and dry specimens.

Material & Methods

The authors have been collecting Bertolonia in Espírito Santo since 2008. Specimens were collected and processed following the usual procedures for botanical specimens (Mori et al., 1989). Morphological descriptions follow Radford et al. (1974), Baumgratz (1983–1985) and Baumgratz (1990). The study was based on literature (Baumgratz, 1990; Baumgratz, Amorim & Jardim, 2011; Bacci et al., 2016) and the analysis of specimens at CEPEC, MBM, MBML, NY, RB, UPCB, VIES (acronyms according to Thiers, 2015). Ecological and geographic distribution data were also obtained from herbarium labels. Conservation status assessments were based on range size (criterion B), according to IUCN Standards and Petitions Subcommittee (2014). The Extension of Occurrence (EOO) and Area of Occupancy (AOO) were calculated through the GeoCAT tool (Bachman et al., 2011). Collection permits were issued by the “Instituto Estadual do Meio Ambiente” (IEMA/Espírito Santo: 14489/07) for the plants inside “Reserva Biológica de Duas Bocas” and by the “Instituto Chico Mendes de Conservação e Biodiversidade (ICMBio: 49043-2) for the plants inside the “Reserva Biológica Augusto Ruschi.”

The electronic version of this article in Portable Document Format (PDF) will represent a published work according to the International Code of Nomenclature for algae, fungi, and plants (ICN), and hence the new names contained in the electronic version are effectively published under that Code from the electronic edition alone. In addition, new names contained in this work which have been issued with identifiers by IPNI will eventually be made available to the Global Names Index. The IPNI LSIDs can be resolved and the associated information viewed through any standard web browser by appending the LSID contained in this publication to the prefix “http://ipni.org/.” The online version of this work is archived and available from the following digital repositories: PeerJ, PubMed Central, and CLOCKSS.

Results

Key to the species of Bertolonia from the state of Espírito Santo, Brazil.

1. Leaf surfaces bullate/foveolate	2	
1′. Leaf surfaces flat	3	
2. Petioles villose; sepals margins entire; petals 5.5−7 mm long	B. foveolata	
2′. Petioles hirsute; sepals margins fimbriate; petals 7.8−8.3 mm long	B. wurdackiana	
3. Leaves abaxial and/or adaxial surfaces permanently covered with sessile and short-stalked glands and long unbranched trichomes	4	
3′. Leaves abaxial and adaxial surfaces covered with only sessile and short-stalked glands	5	
4. Petioles hirsute at the apex; flowers 17−22 mm long; sepals fleshy, with fimbriate margins; petals with entire margins; anther connective with a dorsal appendage	B. formosa	
4′. Petioles glandulose-pilose or glandulose-villose throughout; flowers 11−13 mm long; sepals membranaceous, with entire margins; petals margins ciliate at the apex; anther connective unnapendaged	B. maculata	
5. Hypanthium covered with sessile or short-stalked glands; sepals with an acute apex; petals 6−6.5 mm long, widely obovate	B. ruschiana	
5′. Hypanthium covered with sessile or short-stalked glands and with unbranched trichomes; sepals truncate; petals 7−12 mm, irregularly elliptic	6	
6. Flowers 13.3−14.7 mm long; calyx external teeth ca. 1 mm long, triangular, covered with only sessile and short-stalked glands; petals long-apiculate (ca. 1 mm long), the apiculus without an unbranched trichome at the tip	B. duasbocaensis	
6′. Flowers 9.5−11.2 mm long; calyx external teeth ca. 2 mm long, ovate, covered with long-stalked glands (0.6−0.8 mm long) and sessile and short-stalked glands; petals short-apiculate (ca. 0.5 mm long), the apiculus with an unbranched trichome at the tip	B. macrocalyx	

Bertolonia duasbocaensis Bacci & R.Goldenb. sp. nov.

(Figs. 1 and 2)

Figure 1 Bertolonia duasbocaensis.

(A) Habit. (B) Detail of the abaxial surface of the leaf. (C) Detail of the adaxial surface of the leaf. (D) Abaxial surface of the leaf showing the ciliate margin. (E) Flower bud. (F) Adaxial view of the petal. (G) Detail of the petal apex covered with sessile and short-stalked glands. (H) Lateral view of the stamens. (I) Fruit. (J) Seeds. (A–D) from R.C. Forzza et al. 5035; (E–H) from R. Goldenberg et al. 1249; I and J from R. Goldenberg et al. 1210).

Figure 2 Bertolonia duasbocaensis.

(A) Habit. (B) Infructescence. (C) Adaxial surface of the leaf. (D) Stem and petioles. (E) Inflorescence. (F) Fruit. (G) Flowers on lateral view. (Photos by R. Goldenberg).

Type

Brazil. Espírito Santo: Mun. Cariacica, Reserva Biológica Duas Bocas. Localidade de Alegre, trilha do Pau-Oco. Floresta Ombrófila Densa Montana 20°16′4″S 40°31′30″W 525 m, 18 January 2009, fl., R. Goldenberg, C.N. Fraga, P.H. Labiak, R.C. Forzza, A.P. Fontana, L.C. Kollmann and P.B. Schwartsburd 1249 (Holotype: UPCB!; Isotypes: CEPEC!, MBML!).

Diagnosis

Bertolonia duasbocaensis is similar to B. formosa. The new species has petioles and leaves covered only with sessile or short-stalked glands (vs. petioles hirsute and leaves also covered with unbranched trichomes in B. formosa), smaller flowers (13.3–14.7 mm long vs. 17–22 mm long in B. formosa), inconspicuous and truncate sepals with triangular external teeth (vs. conspicuous, 3.2–3.7 mm long sepals, these rounded and lacking external teeth) and by the petals with the apex covered with sessile or short-stalked glands (vs. petals glabrous).

Description

Herbs 10–35 cm tall, terrestrial, reptant; adventitious roots branched, growing from several points along the stem, but larger next to its base; stem 3–5 mm thick, terete and slightly costate, the older portions plagyotropic and aphyllous, the young ones erect and bearing leaves. Branches, leaves, inflorescences and bracts with sparse to dense, sessile and short-stalked (then less than 0.1 mm long) glands, young branches also with sparse unbranched trichomes. Leaves opposite, occasionally subopposite; petioles 3–9.5 cm long, quadrangular, slightly costate, covered with the same trichomes as the branches; blade 7.8–11.7 × 4.5–9 cm, flat, ovate to elliptic, chartaceous, base rounded to slightly cordate, apex rounded to slightly acute, margins entire to slightly crenulate and ciliate, adaxial surface dark-green, whitish along the primary vein, sparsely covered with sessile glands, otherwise glabrescent, abaxial surface light-green or lilac, sparsely covered with sessile glands, veins three to five, plus a shorter marginal pair that do not reach the leaf apex, basal, main veins at the abaxial surface with pocket domatia at their bases. Thyrsoids 4.5–10.7 cm long, terminal (but pseudo-lateral in older, fruiting specimens), with one pair of paraclades, these cymose, scorpioid, the branches greenish to light-pink; bracteoles 0.5–0.8 mm long, narrow-lanceolate, apex acuminate, both surfaces covered only with sessile glands. Flowers 5-merous, 13.3–14.7 mm long, on pedicels ca. 2 mm long, light green, densely covered with sessile and short-stalked glands. Hypanthium light green, 2.5–2.7 × 2 mm, widely campanulate, 10-costate, with the same indument as the pedicel, seldom with scattered unbranched trichomes in its the upper half. Calyx caducous on fruits, tube ca. 0.7 mm long, sepals truncate, external teeth ca. 1 mm long, green to slightly vinaceous, triangular, apex acute, margins entire, eciliate, both surfaces with the same indument as the pedicels. Petals white, with light-pink or light-purple apex, 10.8–12 × 4.5–5 mm, irregularly elliptic, base slightly atenuate, apex acute and apiculate (ca. 1 mm long), the apiculus bending outwards (to the abaxial surface of the petal), lacking trichomes at its tip, margins entire, both surfaces papillose, adaxial surface apex sparsely covered with sessile and short-stalked glands. Stamens 10, 6–7 mm long, isomorphic; filaments 3.5–4 mm long, slightly widened at the base; thecae yellow, 2–3 mm long, oblong-subulate, slightly undulate, pore apical, introrse; connective prolonged ca. 1 mm below the thecae, dorsally thickened appendage. Ovary free, apex glabrous, 3-locular, placentation axillary; style ca. 5 mm long, curved at the apex, glabrous; stigma slightly capitate, papillose. Capsules bertolonidium-type, 4–5 × 4.5–6 mm, obtriquetrous; seeds 0.4–0.5 mm long, triangular, tuberculate on the dorsal angles and apex.

Distribution and conservation status

Bertolonia duasbocaensis has been collected a few times since 2008, mostly along the same trail (“Pau-Oco”) inside Duas Bocas Biological Reserve, but with one specimen occurring in a private property near the Reserve. The Biological Reserve is a fully protected area, kept by the state government (IEMA/ES). It is located in the municipality of Cariacica, in central Espírito Santo, between 20°14′40′′S–20°18′30′′S and 40°28′01′′W–40°32′07′′W. Its 2,910 ha are covered with well-preserved Atlantic Rainforest that protects threatened species of both fauna and flora (Novelli, 2010). In the last five years, three new plant species were described from Duas Bocas: Eugenia amorimii Fraga & Giaretta (Giaretta & Fraga, 2014), Leandra magnipetala R. Goldenb. & E. Camargo (Camargo & Goldenberg, 2011), and Ouratea cauliflora Fraga & Saavedra (Fraga & Saavedra, 2014). Specimens of Bertolonia duasbocaensis occur in shaded and moist slopes at 500–600 m elev. They were collected with flowers in January and February and with fruits in January, February and July. According to IUCN Standards and Petitions Subcommittee (2014) criteria B1ab(ii) + B2ab(ii), with EOO (Extent of Occurrence) = 2.235 km2 and AOO (Area of Occupancy) = 12.000 km2 , this species should be classified as “critically endangered” (CR).

Paratypes

Brazil. Espírito Santo: Mun. Cariacica, 10 January 2007, fl. and fr., A.P. Fontana et al. 2584 (MBML!, RB!, UPCB!); 10 January 2007, fl., L. Kollmann et al. 9457 (MBML!); 20 July 2008, fr., A.M.A. Amorim et al. 7571 (CEPEC!, RB!, UPCB!); 22 July 2008, fr., R. Goldenberg 1210 (CEPEC!, MBML!, RB!, UPCB!); 15 February 2008, fl. and fr., R.C. Forzza. et al. 5035 (CEPEC!, MBM!, MBML!, RB!, UPCB!). Mun. Viana, Arredores da Reserva Biológica Duas Bocas, São Paulo de Viana, prop. do Valtinho, 700 m, 19 January 2009, fl. and fr., R. Goldenberg et al. 1257 (MBML!, RB!, UPCB!).

Etymology

The epithet “duasbocaensis” refers to the Duas Bocas Biological Reserve, the type locality of the new species.

Remarks

Bertolonia duasbocaensis is characterized by the ovate to elliptic leaves with rounded to slightly cordate bases, margins entire to slightly crenulate and ciliate, 3–5 main veins, and covered only with sessile and short-stalked glands, the widely campanulate hypanthium, truncate sepals, triangular external calyx teeth, apiculate petals with the apex of the adaxial surface covered with sessile and short-stalked glands, and oblong-subulate stamens with a dorsally thickened appendage. The new species closely resembles B. formosa, also endemic of the state of Espírito Santo. For more details on the distinction between the two of them, see the diagnosis above.

Bertolonia hoehneana Brade, B. nymphaeifolia Raddi and B. sanguinea share with the new species the leaves with cordate bases and also the branches and adaxial surface of the leaves covered only with sessile and short-stalked glands. The first is a rare species, endemic to the state of São Paulo (Baumgratz, 1990; Baumgratz, 2016) and can be distinguished by the leaves with a cordate-lobate base (vs. rounded to slightly cordate in B. duasbocaensis), eciliate margins (vs. ciliate margins) and an acute, attenuate, or sometimes acuminate apex (vs. rounded to slightly acute), by the smaller flowers (10–13 mm long vs. 13.5–15 mm long) with the hypanthium covered only with sessile and short-stalked glands (vs. covered also with scattered unbranched trichomes, mostly on its upper half) and sepals widely ovate, lacking external teeth (vs. sepals truncate, with external teeth). Both Bertolonia nymphaeifolia and B. sanguinea are endemic to the state of Rio de Janeiro and differ from the new species by the bigger leaves, with 7–9 and 5–7 acrodromous veins, respectively (vs. 3–5 in B. duasbocaensis), petioles hirsute at the base or seldom on the whole petiole (vs. petioles covered with sessile and stalked glands, the young ones sometimes with caducous, unbranched trichomes), longer flowers, 18–23 mm long in B. sanguinea (vs. shorter ones, 13.3–14.7 mm long in B. duasbocaensis) and by the long subulate stamens with an acute extension at the apex, forming a tube below the apical pore (vs. oblong-subulate stamen, without the extension at the apex). The morphology of the calyx is also different: B. nymphaeifolia has widely ovate, membranaceous sepals with fleshy, rounded external teeth, this forming a concave cavity on the sepals and B. sanguinea has fleshy, widely ovate sepals lacking external teeth (Baumgratz, 1990), while B. duasbocaensis has truncate sepals with a triangular external teeth.

Bertolonia duasbocaensis closely resembles B. macrocalyx; both are endemic of Espírito Santo and occur very close to each other. The differences between both species are listed in the diagnosis of B. macrocalyx (see below).

Bertolonia macrocalyx Bacci & R.Goldenb. sp. nov

(Figs. 3 and 4)

Figure 3 Bertolonia macrocalyx.

(A) Habit. (B) Detail of the adaxial surface of the leaf. (C) Detail of the abaxial surface of the leaf. (D) Adaxial surface of the leaf showing the ciliate margin. (E) Petioles. (F) Flower. (G) Petals of a flower bud showing the sesille and short-stalked glands on the upper half. (H) Frontal view of the calyx. (I) Lateral view of the stamens. (J) Fruit. (A–I from R. Goldenberg et al. 1235; J from C.N. Fraga et al. 2049).

Figure 4 Bertolonia macrocalyx.

(A) Habit. (B) Inflorescence. (C) Frontal view of the calyx and ovary apex. (D) Lateral view of old flowers. (E) Lateral view of the flowers. (F) Frontal view of the flowers. (G) Stamens (Photos by CN Fraga).

Type

Brazil. Espírito Santo: Mun. Viana, Arredores da Reserva Biológica Duas Bocas. São Paulo de Viana, beira da estrada. Floresta Ombrófila Densa Montana 20°18′3″S 40°32′48″W 575 m, 17 January 2009, fl., R. Goldenberg, C.N. Fraga, P.H. Labiak, R.C. Forzza, A.P. Fontana, and P.B. Schwartsburd 1235 (Holotype: UPCB!; Isotypes: CEPEC!, MBML!).

Diagnosis

Bertolonia macrocalyx is similar to B. duasbocaensis. Both species have similar vegetative morphology, but B. macrocalyx has the calyx with longer (ca. 2 mm long), ovate and ciliate external teeth (vs. shorter, ca. 1 mm long, triangular and eciliate in B. duasbocaensis), short-terete hypanthium (vs. widely campanulate hypanthium), smaller, 7–8.5 × 3–3.5 mm petals with the upper half of the adaxial surface densely covered with sessile and short-stalked glands (vs. bigger 10.8–12 × 4.5–5 mm petals, with only the apex of the adaxial surface sparsely covered with sessile and short-stalked glands).

Description

Herbs ca. 20 cm tall, terrestrial or rupicolous, reptant; adventitious roots branched, growing from several points along the stem, but larger next to its base; stem ca. 3 mm thick, quadrangular and slightly costate, the older portions plagyotropic and aphyllous, the young ones erect and bearing leaves. Branches, leaves, inflorescences and bracts with sparse to dense, sessile and short-stalked (then less than 0.1 mm long) glands. Leaves opposite, occasionally subopposite; petioles 1.5–5.3 cm long, quadrangular, slightly costate, covered with the same trichomes as the branches, also with unbranched trichomes, denser at the apex; blade 10–13.5 × 4.7–6.3 cm, flat, ovate, narrowly elliptic to lanceolate, chartaceous, base slightly cordate, apex rounded to slightly acute, margins entire to slightly crenulate and ciliate, adaxial surface dark-green, sometimes light green along the primary vein, abaxial surface light-green or lilac, veins three to five, plus a shorter marginal pair that do not reach the leaf apex, basal, main veins at the abaxial surface with pocket domatia at their bases. Thyrsoids 11.3–16.2 cm long, terminal (but pseudo-lateral in older, fruiting specimens), with one pair of paraclades, these cymose, scorpioid, the branches greenish to light-pink; bracteoles ca. 0.5 mm long, linear to narrow-lanceolate, apex aristate and tipped with one glandular trichome ca. 0.5 mm long, both surfaces covered only with sessile glands. Flowers 5-merous, 9.5–11.2 mm long, on pedicels 1.5–2 mm long, light pink, densely covered with sessile and short-stalked glands. Hypanthium light green, 2.5–2.7 × 2 mm, short-terete, 10-costate, ridges pink, the ones between the external teeth ending in one trichome, with the same indument as the pedicel, also with unbranched trichomes, on the ridges at the apex of the hypanthium. Calyx caducous on fruits, tube ca. 0.8 mm long, sepals truncate, external teeth ca.2 mm long, with the abaxial surface pinkish, abaxial light green, covered with the same trichomes as the pedicels and also with unbranched trichomes (0.6–0.8 mm long) on the adaxial surface, ovate, apex rounded, margins entire, ciliate at the apex. Petals white, with light-pink apex, 7–8.5 × 3–3.5 mm, irregularly elliptic, base slightly atenuate, apex acute and apiculate (ca. 0.5 mm long), the apiculus bending outwards (to the abaxial surface of the petal), ending in a long unbranched trichome, margins entire, both surfaces papillose, upper half of the adaxial surface densely covered with sessile and short-stalked glands. Stamens ten, 6–6.5 mm long, isomorphic; filaments 3.5–4 mm long, slightly widened at the base; thecae yellow, 2–2.5 mm long, oblong-subulate, slightly undulate, pore apical, introrse; connective prolonged ca. 1.5 mm below the thecae, with a dorsal calcar, ca. 0.3 mm long. Ovary free, apex glabrous, 3-locular, placentation axillary; style ca. 5 mm long, curved at the apex, glabrous; stigma slightly capitate, papillose. Capsules bertolonidium-type (following Baumgratz, 1983–1985), 3.5–5 × 4.5–5.3 mm, obtriquetrous; seeds unknown.

Distribution and conservation status

Bertolonia macrocalyx is endemic to the state of Espírito Santo. It had been collected only twice, both in 2008. The species occurs near the limits of the Duas Bocas Biological Reserve, in the municipalities of Cariacica and Viana. The area is covered with well-preserved Atlantic Rainforest. Bertolonia macrocalyx was collected near a local roadside and in a private property. It occurs in shaded and moist slopes, with flowers in January and with old fruits in January, May and June. According to IUCN Standards and Petitions Subcommittee (2014) criteria B1ab(ii) + B2ab(ii), with EOO (Extent of Occurrence) = 0 km2 and AOO (Area of Occupancy) = 8.000 km2 , this species should be classified as “critically endangered” (CR).

Paratype

Brazil. Espírito Santo: Mun. Cariacica, Reserva Biológica Duas Bocas, São Paulo Viana, beira da estrada 575 m, 06 May 2008, fr., C.N. Fraga et al. 2049 (CEPEC!, MBML!, UPCB!).

Etymology

The epithet “macrocalyx” refers to the calyx with long external teeth, one of the diagnostic features of the new species.

Remarks

Bertolonia macrocalyx is characterized by the short petioles (1.5–5.3 cm long) covered with sessile and short-stalked glands and also with unbranched trichomes at the apex, leaves with 3–5 main veins, both surfaces covered with sessile and short-stalked glands, hypanthium short-terete, light green with pinkish ridges, calyx with truncate sepals and external teeth light green on the abaxial surface, ovate and ciliate (trichomes mostly at the apex), petals with the upper half densely covered with sessile and short-stalked glands, short-apiculate, with the apiculus ending in an unbranched trichome, and stamens oblong-subulate, with a dorsally thickened connective appendage.

Bertolonia duasbocaensis also has petals with the apex covered with sessile and short-stalked glands, but they are concentrated at the apex of the adaxial surface and are much more sparsely distributed when compared with B. macrocalyx. Another diagnostic feature of B. macrocalyx is the morphology of the calyx: The sepals are truncate (like B. duasbocaensis), but with a long (ca. 2 mm), ovate and ciliate external teeth. For more details, see the comments above on Bertolonia duasbocaensis

Bertolonia ruschiana Bacci & R.Goldenb. sp. nov

(Figs. 5 and 6)

Figure 5 Bertolonia ruschiana.

(A) Habit. (B) Detail of the abaxial surface of the leaf. (C) Detail of the abaxial surface of the leaf showing the ciliate margin. (D) Hypanthium, calyx and style. (E) Lateral view of the stamens. (F) Flowers and flower bud. (G) Detail of the petals apex covered with sessile and short-stalked glands. (H) Adaxial view of the petal. (I) Fruit. (J) Seeds. (A, I and J from L. Kollmann and E. Bausen 4553; B–H from L.F. Bacci et al. 321).

Figure 6 Bertolonia ruschiana.

(A) Habit. (B) Petiole. (C) Abaxial surface of the leaf blade. (D) Stem. (E) Flowers and flower buds. (F) Inflorescence. (G) Infructescense. (Photos by L.F. Bacci).

Type

Brazil. Espírito Santo: Mun. Santa Teresa, Reserva Biológica Augusto Ruschi. Beira da estrada para Goiapaba-Açu, após a entrada da Reserva. Floresta Ombrófila Densa Montana 19°52′38′′S 40°32′20′′W 800 m, 14 January 2016, fl. and fr., L.F. Bacci, T. Bochorny and M. Bolson 321 (Holotype: UEC!; Isotypes: CEPEC!, MBML!, NY!, UPCB!).

Diagnosis

Bertolonia ruschiana is similar to B. nymphaeifolia. Both have branches, adaxial suface of the leaves, pedicels and hypanthium covered only with sessile and short-stalked glands, white petals with a pinkish apex, hypanthium widely campanulate and a fleshy calyx with widely ovate sepals. The new species differs by the leaves with 5 main acrodromous veins (vs. 7–9 main acrodromous veins in B. nymphaeifolia) and with ciliate margins (vs. eciliate margins), petioles covered with sessile and short-stalked glands, seldom with unbranched trichomes (vs. petioles covered with sessile and short-stalked glands, and also hirsute), shorter flowers (8–9 mm long vs. 12–15 mm long), petals apex covered with sessile and short-stalked glands on the adaxial surface (vs. petals apex glabrous) and ovary apex glabrous (vs. ovary apex covered with glands).

Description

Herbs 10–40 cm tall, terrestrial, seldom rupicolous, reptant; adventitious roots branched, growing from several points along the stem, but larger next to its base; stem 4–7 mm thick, terete and slightly costate, the older portions plagyotropic, aphyllous, the young ones erect and bearing leaves. Branches, leaves, inflorescences and bracts with sparse to dense, sessile and short-stalked (then less than 0.1 mm long) glands. Leaves opposite, occasionally subopposite; petioles 1.5–6.4 cm long, quadrangular, costate, covered with the same trichomes as the branches, seldom with scattered unbranched trichomes at the apex; blade 9–18.5 × 4.5–10.5 cm, flat, ovate, elliptic to wide elliptic, chartaceous, base slightly cordate, apex rounded to slightly acute, sometimes mucronulate, margins crenulate and ciliate, adaxial surface dark-green or with a whitish stripe along the primary vein, covered with sessile glands, abaxial surface light-green or lilac, covered with sessile and short-stalked glands, seldom with small unbranched trichomes on the marginal veins, veins five, plus a shorter marginal pair that do not reach the leaf apex, basal, main veins at the abaxial surface strongly marked. Thyrsoids 3.2–9.5 cm long, terminal (but pseudo-lateral in older, fruiting specimens), with one pair of paraclades, these cymose, scorpioid, the branches greenish to light-brownish; bracteoles 0.5–0.6 mm long, narrow-triangular, apex acuminate, both surfaces covered only with sessile glands. Flowers 5-merous, 8–9 mm long, on pedicels 1.5–2 mm long, densely covered with sessile and short-stalked glands. Hypanthium light green, 2–2.5 × 2 mm, widely campanulate, 10-costate, with the same trichomes as in the pedicel. Calyx caducous, tube ca. 1 mm long, sepals ca. 1.5 mm long, widely ovate, margins entire, both surfaces covered with the same trichomes as in the pedicels, seldom with an unbranched trichome on the tip of the ridges between the sepals, external teeth 0.5–0.7 mm long, light pink, fleshy, triangular, slightly concave, sometimes forming a small cavity, with an apex ending in a small trichome, caducous. Petals white, with a light-pink apex, 6–6.5 × 5.7–6 mm, widely obovate, slightly convex, fleshy, base slightly atenuate, apex rounded to obcordate, margins entire, both surfaces papillose, adaxial surface apex covered sparsely with sessile and short-stalked glands. Stamens ten, 6–7.5 mm long, isomorphic; filaments 3.5–4.5 mm long, papillose, slightly widened at the base; thecae yellow, 2.5–3 mm long, oblong-subulate, slightly undulate, pore apical, introrse; connective prolonged 0.7–0.9 mm below the thecae, dorsally thickened (0.3–0.5 mm long). Ovary free, apex glabrous, 3-locular, placentation axillary; style ca. 6 mm long, curved at the apex, glabrous; stigma slightly capitate, papillose. Capsules bertolonidium-type, 5.5–7 × 6–7.5 mm; seeds 0.5–0.7 mm long, triangular, tuberculate at the apex.

Distribution and conservation status

Bertolonia ruschiana occurs in the central region of the state of Espírito Santo, in three neighboring municipalities: Santa Leopoldina, Santa Maria de Jetibá and Santa Teresa (Fig. 7). Most populations were found in Santa Teresa, within three different conservation units and in several private properties. Two populations with several individuals were found inside the Augusto Ruschi Biological Reserve, a Brazilian Federal Conservation Unit with 3,598 ha (IBAMA–Instituto Brasileiro do Meio Ambiente e Recursos Renováveis, 2000). A few specimens of B. ruschiana were collected in some trails inside the Santa Lúcia Ecological Station and in the São Lourenço Municipal Natural Monument (also called “Caixa D’água”), kept by the Museu de Biologia Professor Mello Leitão (MBML). The first has 400 ha and the second has 363 ha, both covered with Atlantic Rainforest (Mendes & Padovan, 2000). Specimens of B. ruschiana usually are terrestrial or rupicolous herbs that inhabit moist and shaded slopes or rocks with shallow soil near water, but sometimes they are epiphytes growing on lower portions of tree trunks. It was collected with flowers from January to March and with fruits all year. According to IUCN Standards and Petitions Subcommittee (2014) criteria B1ab(ii) + B2ab(ii), with EOO (Extent of Occurrence) = 298.881 km2 and AOO (Area of Occupancy) = 68.000 km2 , this species should be classified as “endangered” (EN). It has been collected more than 30 times, inside three conservation units and several private properties. Nevertheless, its habitat is fragmented and the conservation of the species depends on the maintenance of these units.

Figure 7 Map with the collection localities of the new species: (A) South America, the state of Espírito Santo in black; (B) Detail of the state of Espírito Santo and the neighboring states, the collection points in black; (C) Collection points.

Conservation Units: 1- Duas Bocas Biological Reserve, 2- Santa Lúcia Ecological Reserve, 3- São Lourenço Natural Municipal Reserve, 4- Augusto Ruschi Biological Reserve.

Paratypes

Brazil. Espírito Santo: Mun. Santa Leopoldina: Santo Antônio, 24 October 1988, fr., V. Krause s.n. (MBML 5295!); Bragança, Mata do Tyrol, prop.: Assunta Salvador, 31 October 2006, fr., L.F.S. Magnago 1523 (MBML!); Califórnia, prop. de Albertino Kringer, 16 March 2007, fr., A.P. Fontana et al. 3050 (MBML!, RB!, UPCB!). Mun. Santa Maria de Jetibá: São José do Rio Claro, terreno de L. Butke, 06 May 2001, fr., L. Kollmann et al. 3650 (MBML!). Mun. Santa Teresa: Estação Biológica da Caixa D’água, 11 April 1985, fr., W. Boone 333 (MBML!); Penha, 01 October 1985, fr., H.Q.B. Fernandes 1527 (MBML!); Cabeceira do Rio Saltinho, 03 July 1986, fr., H.Q.B. Fernandes 2003 (MBML!); Valão de São Lourenço, Estação Ecológica da Caixa D’água, 23 December 1993, fr., E. Bausen 53 (MBML!, SPF!, UPCB!); Valão de São Lourenço, Estação Ecológica da Caixa D’água, 07 January 1994, fl. and fr., E. Bausen 54 (MBML!, UPCB!, SPF!); Estação Biológica de Santa Lúcia, 17 January 1995, fr., C.C. Chamas and R.R. Santos 375 (MBML!); Estação Biológica de Santa Lúcia , 24 January 1995, fl., C.C. Chamas 380 (MBML!); 05 March 1999, fl. and fr., L. Kollmann et al. 2039 (MBML!); Estação Biológica de Santa Lúcia, 19 April 2000, fr., L. Kollmann et al. 2869 (MBML!); Valsugana Velha, Prop. Dr. Pedro, 06 June 2001, fr., L. Kollmann 3816 (MBML);Valsugana Velha, 23 June 2001, fr., A.P. Fontana et al. 132 (MBML!); Valsugana Velha, Estação Biológica de Santa Lúcia, 14 September 2001, fr., L. Kollmann and E. Bausen 4553 (MBML!); Reserva Biológica de Nova Lombardia, 05 February 2002, fl. and fr., L. Kollmann et al. 5499 (MBML!, UPCB!); Reserva Biológica Augusto Ruschi, 22 January 2003, fl., R.R. Vervloet et al. 1718 (MBML!); Nova Lombardia, Reserva Biológica Augusto Ruschi, estrada Marlene, 05 February 2003, fl., R.R. Vervloet et al. 1777 (MBML!, RB!, UPCB!); APP São Lourenço, Trilha Boa, 29 March 2003, fr., A.P. Fontana et al. 545 (MBML!, UPCB!); Nova Lombardia, Reserva Biológica Augusto Ruschi, trilha seguindo o córrego, 01 April 2003, fr., R.R. Vervloet and E. Bausen 2095 (MBML!); Reserva Biológica Augusto Ruschi, trilha da Cachoeira, 20 January 2005, fl., H.Q.H. Fernandes and R. Goldenberg 3373 (MBML!, UPCB!); Santo Henrique, 26 January 2005, fl., L. Kollmann 7308 (MBML!); Santo Henrique, beira da estrada, 26 January 2005, fl., L. Kollmann and A.P. Fontana 7328 (MBML!, UPCB!); Penha, prop. sr. Tabajara Ribeiro de Oliveira, 22 May 2005, fr., R.C. Britto 58 (MBML!); Santo Anselmo, terreno do M. Nandolfo, 830 m, 24 February 2006, fr., L. Kollmann et al. 8675 (MBML!); Julião, prop. Sr. João Luiz Rodrigues de Souza, 23 February 2007, fr., A.P. Fontana 2980 (MBML!); Nova Lombardia, Terreno do Furlani, 13 July 2007, fr., R. Goldenberg et al. 891 (MBML!, NY!, UPCB!); Lombardia, prop. de João Furlane, 19°48′9′′S 40°32′14′′W, 800 m, 4 May 2009, fr., A.P. Fontana et al. 5982 (MBML!, WAG); Nova Lombardia, terreno do Furlani, 7 February 2011, fr. and fl., F.A. Michelangeli et al. 1604 (NY!; UPCB!); Nova Lombardia, terreno do Furlani, 7 February 2011, fl., F.A. Michelangeli et al. 1606 (NY!; UPCB!); Reserva Biológica Augusto Ruschi, 10 December 2012, fr., J.A. Lombardi 9819 (UPCB!); Reserva Biológica Augusto Ruschi, 27 November 2013, fr., L.F. Bacci and D.F. Lima 108 (UPCB!); Reserva Biológica Augusto Ruschi, trilha da Roda D’água, 14 January 2016, fl. and fr., L.F. Bacci et al. 317 (MBML!, NY!, UEC!, UPCB!).

Etymology

The epithet “ruschiana” honors the Brazilian naturalist Augusto Ruschi (1915–1986), who fought against the deforestation of the Atlantic Forest, mainly in the state of Espírito Santo. In addition, the type locality of the species is in the Augusto Ruschi Biological Reserve.

Remarks

Bertolonia ruschiana is characterized by the adaxial surface of the leaves, pedicels and hypanthium covered only with sessile and short-stalked glands, the leaves with 5 main acrodromous veins and ciliate margins, hypanthium widely campanulate, sepals widely ovate and fleshy, with triangular, slightly concave external teeth. The widely depressed, slightly convex and fleshy petals with the apex of the adaxial surface covered sparsely with sessile and short-stalked glands are also important for the recognition of the new species. Bertolonia ruschiana closely resembles B. nymphaeifolia (for more details, see the diagnosis above).

Several specimens of Bertolonia ruschiana had been determined as Bertolonia sanguinea, due to the similar vegetative morphology. Both species share the big leaves with cordate bases, and both surfaces of the leaf covered only with sessile or short-stalked glands, but B. sanguinea has hirsute petioles and eciliate leaf margins (Baumgratz, 1990). The morphology of the flowers is also different. Bertolonia sanguinea has sepals lacking external teeth (vs. sepals with a triangular, concave external teeth in B. ruschiana), longer flowers (18–23 mm vs. 8–9 mm), glabrous petals (vs. adaxial surface apex covered with sessile and short-stalked glands), longer stamens (18–19 mm long vs. 6–7.5 mm), these with an extension at the apex, forming a tube (vs. stamens without an extension at the apex) (Fig. 7).

The authors would like to thank Cláudio N. Fraga for the photos, Paulo Labiak, Rafaela Forzza, Ludovic Kollmann, André Fontana, Thuane Bochorny and Mônica Bolson for the help in the field, and also Helio B.Q. Fernandes and the staff at MBML (Museu de Biologia Mello-Leitão) for the support in the field and for providing accommodation at Santa Lucia. We also would like to thank Fabián A. Michelangeli and Paulo J. Guimarães for the careful review of the manuscript.

Additional Information and Declarations

Competing Interests

Author Contributions

Field Study Permissions

Data Availability

New Species Registration

The authors declare there are no competing interests.

Lucas F. Bacci conceived and designed the experiments, performed the experiments, analyzed the data, wrote the paper, prepared figures and/or tables, reviewed drafts of the paper.

André M. Amorim and Renato Goldenberg conceived and designed the experiments, analyzed the data, wrote the paper, reviewed drafts of the paper.

The following information was supplied relating to field study approvals (i.e., approving body and any reference numbers):

Collection permits for this study were issued by IEMA/ES #14489/07 and ICMBio 49043-2.

The following information was supplied regarding data availability:

The raw data is included in the figures.

The following information was supplied regarding the registration of a newly described species:

1334577158860-1 - _Bertolonia duasbocaensis, _77158861-1 - _Bertolonia macrocalyx, _77158862-1 - Bertolonia ruschiana.

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
