# Peer review of "Three new species of Bertolonia (Melastomataceae) from Espírito Santo, Brazil"

_PeerJ, doi:10.7717/peerj.2822_

## Round 0.1 · original submission · Minor Revisions

Please pay particular attention to the comments by the reviewers on the annotated files, particularly those indicated by Reviewer #1.

·

Basic reporting

In this manuscript Bacci and collaborators describe three new species of Bertolonia, all from the state of Espiritu Santo in Eastern Brazil. Bertolonia is endemic to Eastern Brazil and ususally restricted to Atlantic Forests. This manuscript underscores large gaps in our knowledge of the biodiversity of this biome.
The manuscript is well written and the species descriptions accurate and clear. The illustrations provide great additional information.

Experimental design

My only comments are regarding the key. I think that some of the couplets can be improved by placing the entire organ before the characters, more like it would be written in a description than in prose. I have made several of these changes already, but the authors should check for consistency throughout.
Also, the AOO and EOO are provided for the conservation assessments of first species, only AOO for the second and neither for B. ruschiana. Since the methods talk about AOO and EOO, and in the interest of consistency, these measurements should be provided for all three species.
A couple of comments on terminology.
The authors use the term laciniate for describing the sepal margins. I think that because laciniate means divided into deep narrow irregular segments it is hard to separate whether they are describing the entire sepal or just the margin; I would prefer to use the term fimbriate.
Also, the term eciliate is used for describing the margin of the petals in some species. As I prefer to use terms that describe the plant for what it is and not for what it is not, I think that margin entire (or equivalent as necessary) would be better.

Validity of the findings

See above

Additional comments

I am attaching the manuscript with some of these and additional comments and minor edits.

·

Basic reporting

"No Comments".

Experimental design

"No Comments".

Validity of the findings

"No Comments".

Additional comments

The manuscript is well prepared and this is the result of an effort to knowledge of capixaba flora. I congratulate the fieldwork that was fundamental for the characterization of species and determine the degree of threat. A few corrections are highlighted in the manuscript.

---

## Round 0.2 · accepted · Accept

I believe that the modifications on the revised version fully address the points raised by the reviewers.